# Applications to medical and surgical specialist training in the UK National Health Service, 2021–2022: a cross-sectional observational study to characterise the diversity of successful applicants

Dinesh Aggarwal  ,[1,2] Meera Roy-Chowdhury,[3] Nicola Xiang,[4] Sharon J. Peacock[1]

[1]Department of Medicine, University of Cambridge, Cambridge, UK
[2]Wellcome Sanger Institute, Hinxton, UK
[3]Jones Lang LaSalle Incorporated (JLL), London, UK
[4]Imperial College Healthcare NHS Trust, London, UK

**Correspondence to**
Dr Dinesh Aggarwal;
dinesh.aggarwal@nhs.net

## ABSTRACT

**Objectives** To compare success of applicants to specialty training posts in the UK by gender, ethnicity and disability status.

**Design** Cross-sectional observational study.

**Setting** National Health Service, UK.

**Participants** All specialty training post applications to Health Education England, UK, during the 2021–2022 recruitment cycle.

**Intervention** Nil.

**Primary and secondary outcome measures** Comparison of success at application to specialty training posts by gender, ethnicity, country of qualification (UK vs non-UK) and disability. The influence of ethnicity on success was investigated using a logistic regression model, where country of qualification was included as a covariate.

**Results** 12 419/37 971 (32.7%) of applicants to specialty training posts were successful, representing 58 specialties. The difference in percentage of successful females (6480/17 523, 37.0%) and males (5625/19 340, 29.1%) was 7.9% (95% CI 6.93% to 8.86%), in favour of females. Segregation of applications to specialties by gender was observed; surgical specialties had the highest proportion of male applicants, while obstetrics and gynaecology had the highest proportion of female applicants. The proportion of successful recruits to specialties largely reflected the number of applications. 11/15 minority ethnic groups (excluding 'not stated') had significantly lower adjusted ORs for success compared with white-British applicants. 'Mixed white and black African' (OR 0.52, 95% CI 0.44 to 0.61, p≤0.001) were the least successful minority group in our study, while non-UK graduates had an adjusted ORs for success of 0.43 (95% CI 0.41 to 0.46, p≤0.001) compared with UK graduates. The difference in percentage of success by disabled applicants (179/464, 38.6%) and non-disabled applicants (11 940/36 418, 32.8%) was 5.79% (95% CI 1.23% to 10.4%), in favour of disabled applicants. No disabled applicants were accepted to 21/58 (36.2%) of specialties.

**Conclusions** Despite greater success by female applicants overall, there is an attraction issue to specialties

## STRENGTHS AND LIMITATIONS OF THIS STUDY

⇒ This study uses a large sample of applicants to a national recruitment portal allowing the estimation of success at application to specialist recruitment posts by demographic groups.

⇒ We evaluate three important protected characteristics—gender, ethnicity and disability—representing the most comprehensive study of the inclusivity of the national recruitment process, to our knowledge.

⇒ By including country of qualification as a covariate in our analysis, we account for a key confounding variable.

⇒ Due to Information Commission Office standards, small specialties are largely excluded from the individual specialty level analysis and intersectional analysis beyond what is presented is not possible.

⇒ Due to limitations on data availability longitudinal effects and those of residual confounding factors, such as the influence of socioeconomic background on the disparate success observed by ethnic minority groups, cannot be ascertained here but are important to study in future work.

by gender. Further, most ethnic minority groups are less successful at application when compared with white-British applicants. This requires continuous monitoring and evaluation of the reasons behind observed differences.

**Trial Registration** Not applicable.

## INTRODUCTION

It is crucial for the National Health Service (NHS) to reflect the society which it serves and to nurture diverse talent effectively. Harnessing diverse lived experiences and perspectives strengthens the pool of knowledge and skills within the profession. Diverse teams are more efficient, innovative and make better decisions, meaning that they are ultimately better placed to serve patients.[1]

Importantly, a lack of workforce diversity can be detrimental to patient care, with research demonstrating inherent biases influencing how clinicians treat patients.[2] The NHS is understaffed at all levels of seniority; backlogs created by the COVID-19 crisis, systemic underfunding, the increasingly complex care needs of an ageing population and issues with retention of its medical workforce are only serving to exacerbate this.[3] The UK is reported to employ 3.03 doctors per 1000 people compared with an average of 3.7 per 1000 people in the Organisation for Economic Co-operation and Development European Union nations.[4] These issues are likely to only worsen as the workforce supply and demand gap is predicted to widen.[5] Creating a more inclusive culture in the NHS is required to ensure that a wider, more diverse talent pool is attracted, able to break into the profession, able to progress successfully and be retained.

Disparate representation within the NHS workforce is well reported.[6–8] Surgical specialties have the lowest proportion of female consultants, and gender parity within surgical specialties is predicted to be reached in decades rather than years.[8] Such disparities have inevitable knock-on effects. A lack of female representation contributes towards a male-dominated culture, and results in fewer female role models to inspire and encourage aspiring female doctors. These are known factors in deterring female applicants as early as undergraduate training.[9] A recent independent report examining the gender pay gap in medicine in England found that the mean whole-time equivalent pay gap is 18.9% for hospital doctors, 15.3% for GPs and 11.5% for clinical academics in favour of men; a key recommendation to reduce gender pay was to minimise gender segregation within specialties.[6] Additionally, 41.9% of the NHS medical workforce comprises individuals from minority ethnic groups, yet only 22.7% at clinical director level.[7] Potential bias against minority ethnic groups embedded in recruitment processes and decisions has been demonstrated in multiple sectors and can occur at multiple stages.[10–12] A historical study from Esmail and Everington reported poorer outcomes for matched applicants to specialty medicine with Asian names compared with those with English names.[10] More recent data have highlighted differences in applicants deemed appointable for specialty posts by ethnicity, but the data failed to account for the impact of country of graduation—a known additional factor in success at postgraduate examinations and recruitment.[13 14]

As the largest employer in the UK, the NHS must have fair and transparent recruitment processes. This responsibility is reinforced by the fact that the NHS is governed by the public sector equality duty with the stated objectives to: 'eliminate unlawful discrimination; advance equality of opportunity between people who share a protected characteristic and those who don't; foster or encourage good relations between people who share a protected characteristic and those who don't'.[15] Systematic examination of disparities in recruitment data by protected characteristics across specialties is, therefore, a key missing piece of analysis. Investment of time and resources in investigating any disparities highlighted, addressing the root causes through action and mitigation, and evaluating and measuring progress over time will help to progress diverse representation throughout all levels of seniority, and create a more inclusive culture across the NHS clinical workforce. Here, we examine disparities in the outcomes for the recruitment to all specialty training posts in the UK in 2021 by gender, ethnicity and disability status.

## METHODS

### Specialty training post recruitment process

Obtaining a national training number for specialty training is the route which generates the vast majority of new consultants and general practitioners in the UK. Depending on the specialty, length of training can be variable and entry points for candidates can also vary (ie, some postgraduate junior doctors may enter specialty training in the first year of the respective specialty training programme or may be deemed to have the competencies required to apply at a later stage in the programme; these entry points are donated by a prefix of 'ST' or 'CT' followed by the year of training at which the specialty training posts is allowing entry. Applications are made online and mostly processed/organised by a lead NHS Health Education England (HEE) local office/devolved nation in order to prevent the need for multiple applications to access posts in different geographical locations. Briefly, the process involves:

► Shortlisting of candidates which can include self-assessment or a multispecialty recruitment assessment.
► An interview, recommended to be multipanel.
► Ranking based on shortlisting and interview scores.
► Offer of appointment.

Demographic details are self-reported. Options for gender include 'male', 'female' and 'not stated'. Options for ethnicity include 'white—British ', 'Asian or Asian British—Bangladeshi', 'Asian or Asian British—Indian', 'Asian or Asian British—Pakistani', 'Chinese', 'any other Asian background', 'black or black British—African', 'black or black British—Caribbean', 'any other black background', 'mixed white and Asian', 'mixed white and black African', 'mixed white and black Caribbean', 'any other mixed background', 'white—Irish', 'any other white background', 'any other ethnic background' and 'not stated'. Options for disability include 'yes' (ie, applicant has a disability), 'no' and 'not stated', and data pertaining to the category of disability was not available for analysis. Further details on the application process for individual specialties in the UK can be found here, https://specialtytraining.hee.nhs.uk/Recruitment.[16]

### Study participants

Data on the gender, ethnicity, disability status and country of qualification (UK vs non-UK) of applicants to specialty

training posts for the 2021–2022 recruitment cycle were made available through two separate Freedom of Information (FOI) requests to HEE. HEE follows the Information Commissioner's Office code of practice relating to data anonymisation; release of small numbers (<5 in a geographical area/demographic) has, therefore, been suppressed. All data made available through the FOI request, including specialties where data are incomplete due to low numbers, can be found in online supplemental data.

## Statistical analysis

Analysis was performed in Excel 2016 and R V.4.02. Figures were generated in R V.4.2.1. Probabilities were calculated from estimated models using the emmeans package (V.1.7.3).[17] Where individual specialties have been grouped for graphical representation, the breakdown of these groups (eg, surgical) can be found in online supplemental table 1. Differences in percentages of successful recruitment between categories are reported with 95% CIs derived using the prop.test function in R. Where 95% CIs do not cross the line of no-effect (ie, zero in the case of differences in percentages of successful recruitment between categories), the result is statistically significant. When assessing completeness of data to consider the difference in percentages of successful applicants by gender, data relating to applicants with undisclosed gender are excluded. The influence of ethnicity on success at application to specialty posts is investigated using a logistic regression model by univariable and multivariable analysis (outcome~graduate status+ethnic origin) where country of qualification (UK vs non-UK) is included as a covariate. Variation of success at application between ethnic groups by country of qualification is then examined by including interaction terms (outcome~graduate status+ethnic origin+graduate status:ethnic origin) between the two variables. This study complies with the STrengthening the Reporting of OBservational studies in Epidemiology (STROBE) reporting guidelines for observational studies in epidemiology.[18]

## Patient and public involvement

Patients or the public were not involved in study design, data collection and analysis, decision to publish, or preparation of the manuscript.

## RESULTS

There were 12 419/37 971 (32.7%) successful applicants to HEE for training posts representing 58 specialty posts in 2021; a distribution of applications by specialty is shown in online supplemental table 2.

## Recruitment by gender

When considering all applicants, we found 17 523/37 971 (46.1%) were female, and 1108/37 971 (2.9%) individuals preferred not to state their gender. Of the successful applicants, 6480/12 419 (52.2%) were female

and 5625/12 419 (45.3%) were male. Overall, the difference in percentage of success by females (6480/17 523, 37.0%) and males (5625/19 340, 29.1%) was 7.9% (95% CI 6.93% to 8.86%), in favour of females (figure 1, online supplemental figure 1).

## Recruitment to specialties by gender

Complete data representing male and female applications were available for 56/58 (96.6%) specialty training posts, while 40/58 (69.0%) specialty training posts had complete data representing successful male and female applications. When considering applications by individual specialty, 13/40 (32.5%) specialty posts had a significantly higher percentage of successful females when compared with males, compared with 2/40 (5.0%) specialty posts that had a higher percentage of successful males (figure 1). Despite this, 22/40 (55.0%) specialty posts had a greater absolute number of successful male applicants compared with females, representing an 'attraction' issue. When examining absolute numbers of applicants and successful applicants by gender, there was an observed segregation by specialty (figure 2, online supplemental figures 2 and 3). Of note, surgical specialties (3097/4742, 65.3%) and radiology (1146/1783, 64.3%) had the highest proportion of male applicants, while obstetrics and gynaecology (957/1321, 72.4%) and public health (644/959, 67.2%) had the highest proportion of female applicants. The gender balance of successful recruits to specialties largely reflected that of the pool of applicants (figure 2C).

## Recruitment by ethnicity and country of qualification

Applications for specialty training posts by ethnicity are shown in figure 3. In the 2021–2022 application cycle, 19 044/37 971 (50.2%) of applicants to specialty training posts were non-UK medical graduates (online supplemental figure 4). Excluding non-medical applicants to Public Health Training Posts, of the successful applicants, 4334/12 419 (34.9%) were non-UK graduates and 7987/12 419 (64.3%) were UK graduates. Overall, the difference in percentage of success by UK graduates (7987/17 939, 44.5%) and non-UK graduates (4334/19 044, 22.8%) was 21.8% (95% CI 20.8% to 22.7%), in favour of UK graduates (online supplemental table 4). UK graduates were significantly more successful in all ethnic groups other than 'Chinese' and 'any other black background' (online supplemental table 4).

Using a multivariable logistic regression analysis, including country of qualification as a covariate, we found 11/15 (73.3%) minority ethnic groups (excluding 'not stated') to have a significantly lower OR of success at recruitment when compared with white-British applicants (figure 4 and online supplemental table 5). Non-UK graduates had a significantly lower OR of success at recruitment when compared with UK graduates (adjusted OR 0.43, 95% CI 0.41 to 0.46, p≤0.001). All adjusted and unadjusted ORs for success by country of qualification and minority ethnic group are provided in online supplemental tables 5 and 6. Additionally, only the

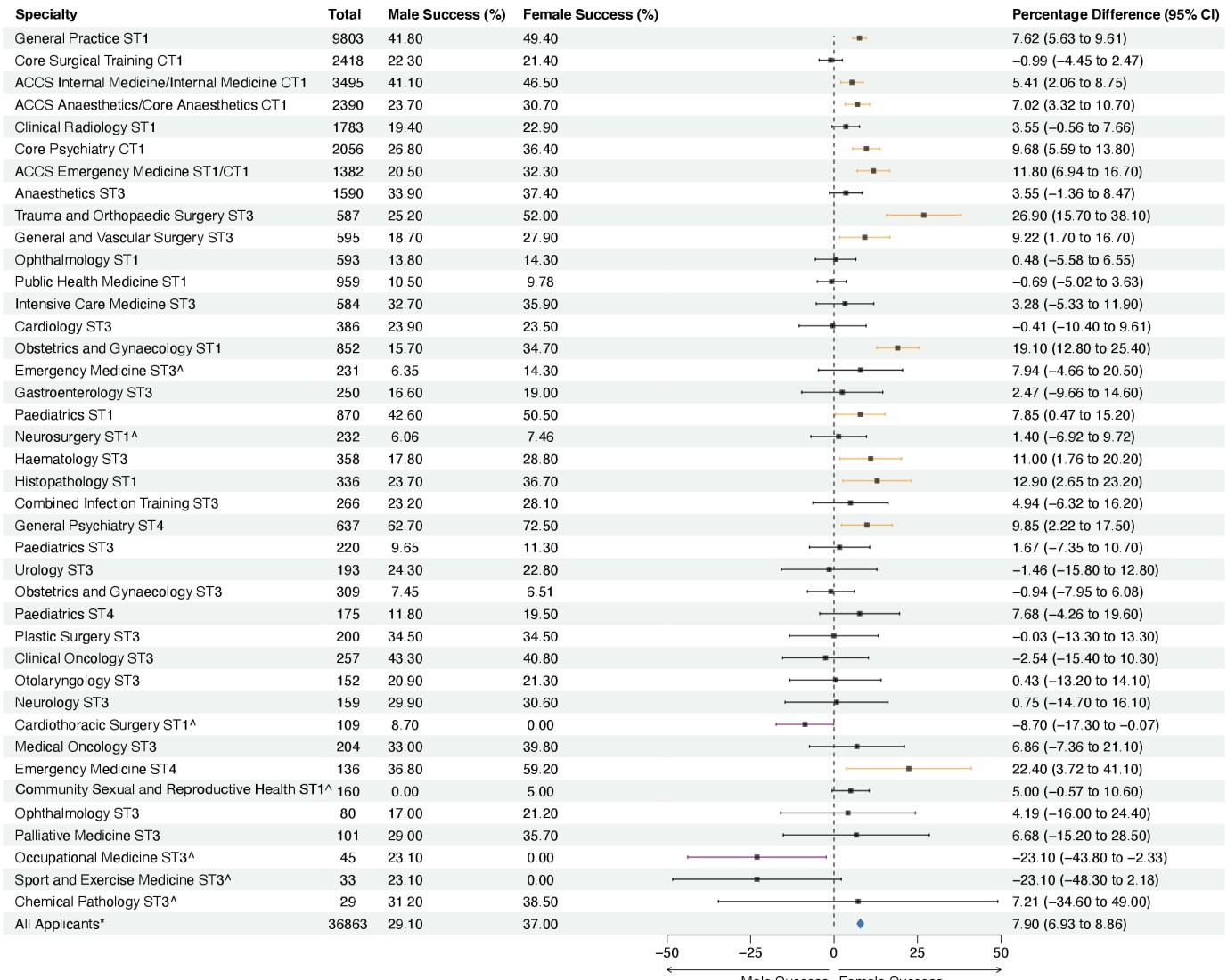

| Specialty | Total | Male Success (%) | Female Success (%) | Percentage Difference (95% CI) |
|---|---|---|---|---|
| General Practice ST1 | 9803 | 41.80 | 49.40 | 7.62 (5.63 to 9.61) |
| Core Surgical Training CT1 | 2418 | 22.30 | 21.40 | −0.99 (−4.45 to 2.47) |
| ACCS Internal Medicine/Internal Medicine CT1 | 3495 | 41.10 | 46.50 | 5.41 (2.06 to 8.75) |
| ACCS Anaesthetics/Core Anaesthetics CT1 | 2390 | 23.70 | 30.70 | 7.02 (3.32 to 10.70) |
| Clinical Radiology ST1 | 1783 | 19.40 | 22.90 | 3.55 (−0.56 to 7.66) |
| Core Psychiatry CT1 | 2056 | 26.80 | 36.40 | 9.68 (5.59 to 13.80) |
| ACCS Emergency Medicine ST1/CT1 | 1382 | 20.50 | 32.30 | 11.80 (6.94 to 16.70) |
| Anaesthetics ST3 | 1590 | 33.90 | 37.40 | 3.55 (−1.36 to 8.47) |
| Trauma and Orthopaedic Surgery ST3 | 587 | 25.20 | 52.00 | 26.90 (15.70 to 38.10) |
| General and Vascular Surgery ST3 | 595 | 18.70 | 27.90 | 9.22 (1.70 to 16.70) |
| Ophthalmology ST1 | 593 | 13.80 | 14.30 | 0.48 (−5.58 to 6.55) |
| Public Health Medicine ST1 | 959 | 10.50 | 9.78 | −0.69 (−5.02 to 3.63) |
| Intensive Care Medicine ST3 | 584 | 32.70 | 35.90 | 3.28 (−5.33 to 11.90) |
| Cardiology ST3 | 386 | 23.90 | 23.50 | −0.41 (−10.40 to 9.61) |
| Obstetrics and Gynaecology ST1 | 852 | 15.70 | 34.70 | 19.10 (12.80 to 25.40) |
| Emergency Medicine ST3^ | 231 | 6.35 | 14.30 | 7.94 (−4.66 to 20.50) |
| Gastroenterology ST3 | 250 | 16.60 | 19.00 | 2.47 (−9.66 to 14.60) |
| Paediatrics ST1 | 870 | 42.60 | 50.50 | 7.85 (0.47 to 15.20) |
| Neurosurgery ST1^ | 232 | 6.06 | 7.46 | 1.40 (−6.92 to 9.72) |
| Haematology ST3 | 358 | 17.80 | 28.80 | 11.00 (1.76 to 20.20) |
| Histopathology ST1 | 336 | 23.70 | 36.70 | 12.90 (2.65 to 23.20) |
| Combined Infection Training ST3 | 266 | 23.20 | 28.10 | 4.94 (−6.32 to 16.20) |
| General Psychiatry ST4 | 637 | 62.70 | 72.50 | 9.85 (2.22 to 17.50) |
| Paediatrics ST3 | 220 | 9.65 | 11.30 | 1.67 (−7.35 to 10.70) |
| Urology ST3 | 193 | 24.30 | 22.80 | −1.46 (−15.80 to 12.80) |
| Obstetrics and Gynaecology ST3 | 309 | 7.45 | 6.51 | −0.94 (−7.95 to 6.08) |
| Paediatrics ST4 | 175 | 11.80 | 19.50 | 7.68 (−4.26 to 19.60) |
| Plastic Surgery ST3 | 200 | 34.50 | 34.50 | −0.03 (−13.30 to 13.30) |
| Clinical Oncology ST3 | 257 | 43.30 | 40.80 | −2.54 (−15.40 to 10.30) |
| Otolaryngology ST3 | 152 | 20.90 | 21.30 | 0.43 (−13.20 to 14.10) |
| Neurology ST3 | 159 | 29.90 | 30.60 | 0.75 (−14.70 to 16.10) |
| Cardiothoracic Surgery ST1^ | 109 | 8.70 | 0.00 | −8.70 (−17.30 to −0.07) |
| Medical Oncology ST3 | 204 | 33.00 | 39.80 | 6.86 (−7.36 to 21.10) |
| Emergency Medicine ST4 | 136 | 36.80 | 59.20 | 22.40 (3.72 to 41.10) |
| Community Sexual and Reproductive Health ST1^ | 160 | 0.00 | 5.00 | 5.00 (−0.57 to 10.60) |
| Ophthalmology ST3 | 80 | 17.00 | 21.20 | 4.19 (−16.00 to 24.40) |
| Palliative Medicine ST3 | 101 | 29.00 | 35.70 | 6.68 (−15.20 to 28.50) |
| Occupational Medicine ST3^ | 45 | 23.10 | 0.00 | −23.10 (−43.80 to −2.33) |
| Sport and Exercise Medicine ST3^ | 33 | 23.10 | 0.00 | −23.10 (−48.30 to 2.18) |
| Chemical Pathology ST3^ | 29 | 31.20 | 38.50 | 7.21 (−34.60 to 49.00) |
| All Applicants* | 36863 | 29.10 | 37.00 | 7.90 (6.93 to 8.86) |

−50   −25   0   25   50
← Male Success   Female Success →

**Figure 1** Success of applicants to training posts by specialty in the 2021 recruitment year by gender. Figure showing the applications of trainees to specialty training posts and percentage of successful applicants by gender, where data are complete. All values are provided to three significant figures. Where CIs for estimated differences between the percentage of successful applicants by gender does not cross zero, bars are highlighted in orange (greater female success) or purple (greater male success). Specialties marked with ∧contained small numbers (ie, expected frequencies were lower than five in any domain of the contingency table) and therefore Fisher's exact test results to examine differences in proportions of successful candidates are also provided in online supplemental table 3.

'Asian or Asian British—Bangladeshi' group significantly differed in outcome across the graduate status (p=0.02) in the interaction model (online supplemental table 7 and figure 5).

**Recruitment by disability**

When considering all applicants to specialty training posts, we found 464/37 971 (1.2%) were disabled, and 1089/37 971 (2.9%) individuals preferred not to state their disability status. Of the successful applicants, 179/12 419 (1.4%) were disabled and 11 940/12 419 (96.1%) were non-disabled. Overall, the difference in percentage of success by disabled applicants (179/464, 38.6%) and non-disabled applicants (11 940/36 418, 32.8%) was 5.79% (95% CI 1.23% to 10.4%), in favour of disabled applicants (online supplemental figure 6). However,

there were no disabled applicants to 13/58 (22.4%) specialties and a further 21/58 (36.2%) specialties where no disabled applicants were accepted. Of the specialties where data were available, general psychiatry ST4 had the highest acceptance rate for disabled applicants (15/16, 93.8%), while general practice had the highest absolute number of successful applicants (61/121, 50.4%).

**DISCUSSION**

We present the most complete assessment of diversity data relating to recruitment to specialty training posts in the UK to our knowledge. Overall, of the specialties studied, female applicants were more successful in the recruitment process though we note clear segregation

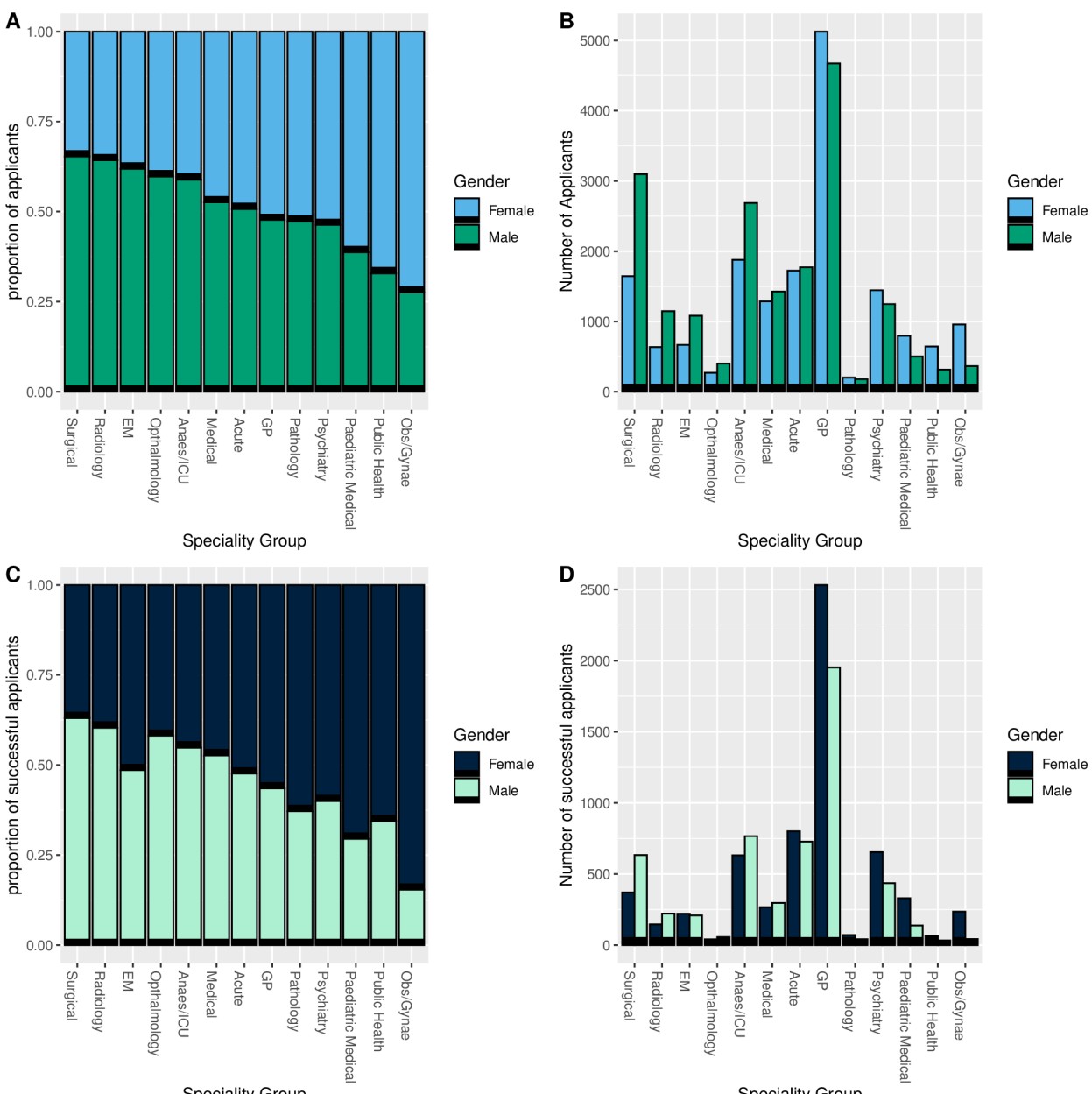

**Figure 2** Applications and outcomes for specialty training posts separated into groups of specialties in the 2021–2022 recruitment year by gender. (A) Proportion of applicants to specialty groups by gender. (B) Number of applicants to specialty groups by gender. (C) Proportion of successful applicants to specialty groups by gender. (D) Number of successful applicants to specialty groups by gender. Groupings of specialties can be found in online supplemental table 1. EM, Emergency Medicine; ICU, Intensive Care Unit; GP, general practitioner.

of applications by specialty. We observe a low absolute number of successful female applicants to surgical specialties, radiology and ophthalmology, and a low absolute number of successful male applicants to obstetrics and gynaecology, public health and paediatric medicine. We show that several minority ethnic groups are less likely to be successful in their applications when adjusting for country of graduation. Finally, we find disabled applicants to be more successful in the recruitment process when compared with non-disabled applicants, but again with significant variation in applications and success at individual specialty level. The NHS has committed to fostering a truly inclusive environment; this work, and the

associated actions (box 1), should serve as a framework to evaluate recruitment data, and subsequently review policies and processes to enable the implementation of improvements, to ensure that this commitment can be achieved.

### Strengths and limitations of this study, and future work

We use a large sample of applicants to a national recruitment portal maintained by HEE, which allows us to estimate the success at application to specialist recruitment posts by demographic groups. We evaluate three important protected characteristics, gender, ethnicity and disability, which represents the most comprehensive

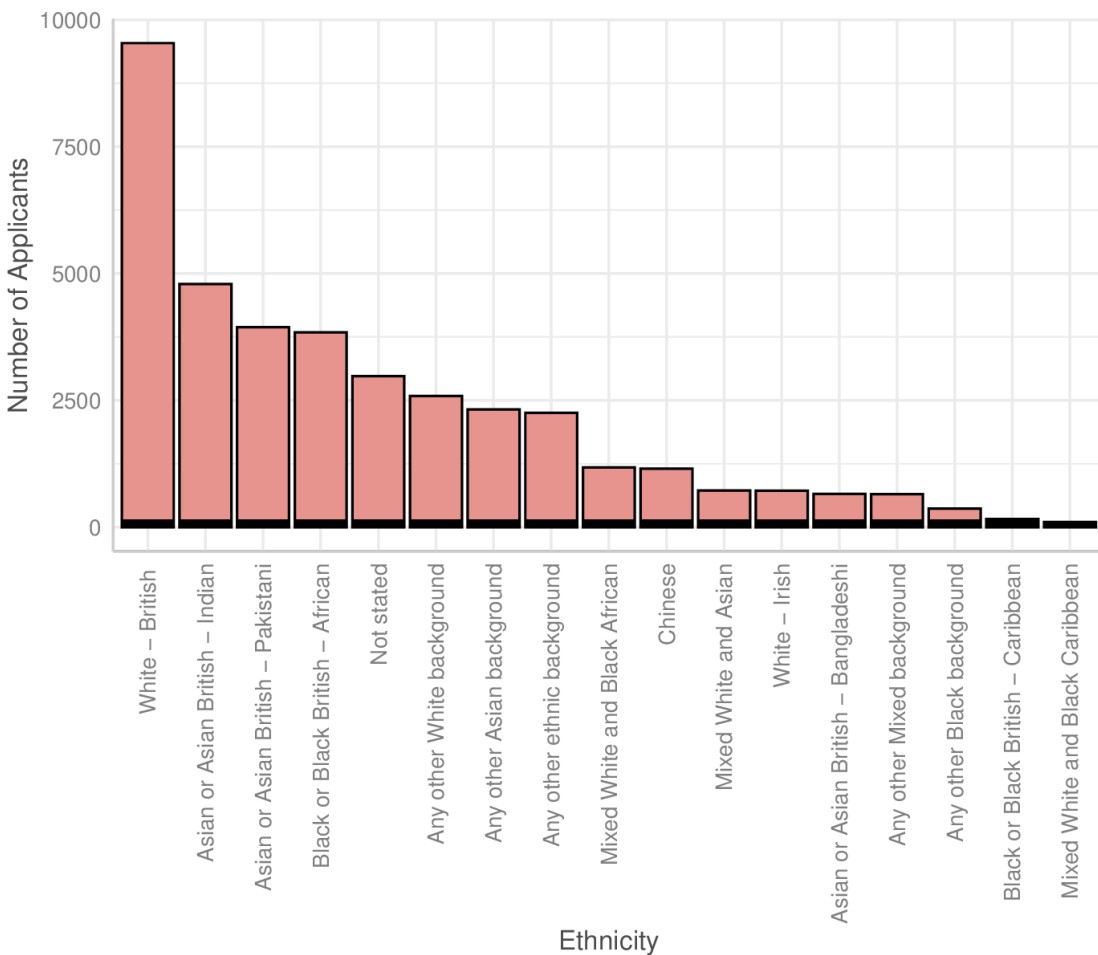

**Figure 3** Applications and outcomes for specialty training posts in the 2021–2022 recruitment year by ethnicity. Applications to specialty training posts in the 2021 recruitment year by ethnicity.

evaluation of the inclusivity of the national recruitment process to our knowledge. By including country of qualification as a covariate in our analysis, we account for a key confounding variable, demonstrating unequal success by ethnicity despite country of qualification; an important result that suggests the recruitment process needs further investigation.

This study has a number of limitations. Due to Information Commission Office standards, small specialties are largely excluded from the individual specialty level analysis. Data are particularly incomplete at individual specialty level for ethnicity and disability. Further, a comparative analysis by specialty is limited by the small numbers in each specialty and beyond the scope of this study. Due to the nature of the data, multivariable analysis beyond what is presented is not possible; an intersectional analysis of protected characteristics, for example, females from ethnic minority backgrounds, should be conducted to identify groups that may be particularly disadvantaged. Further, residual confounding factors, such as the influence of socioeconomic background on the disparate success observed by ethnic minority groups cannot be ascertained here but present an important factor to study in future work. The data are representative of 1 year of recruitment, and possible enduring effects of

the COVID-19 pandemic on applications and recruitment patterns cannot be disregarded; ongoing systematic data monitoring is necessary to evaluate trends in applications and successful recruitment by demographic characteristics over time. The data were obtained through two separate FOI requests; one representing aggregated data for 'all specialty training posts' and the second including data separated by individual specialties. Data analysis does not include comparison of the aggregated data between datasets but we do note a minor difference in the total number of applicants between datasets (raw data are provided in online supplemental files)—given the large sample size, these are not expected to influence our findings. We report an analysis of recruitment to specialty training posts which is only one aspect of assessing the diversity and inclusiveness of a profession; additional work should assess how these findings relate to pay gaps, progression and retention of the workforce, and for any barriers faced by groups representing other protected characteristics as defined by the Equality Act 2010[15] within the NHS. We note 'black or black British—African', 'black or black British—Caribbean', and 'mixed white and black Caribbean' applicants do not have a lower OR of success at application when compared with white British applicants; this does not mean the absolute numbers of applicants

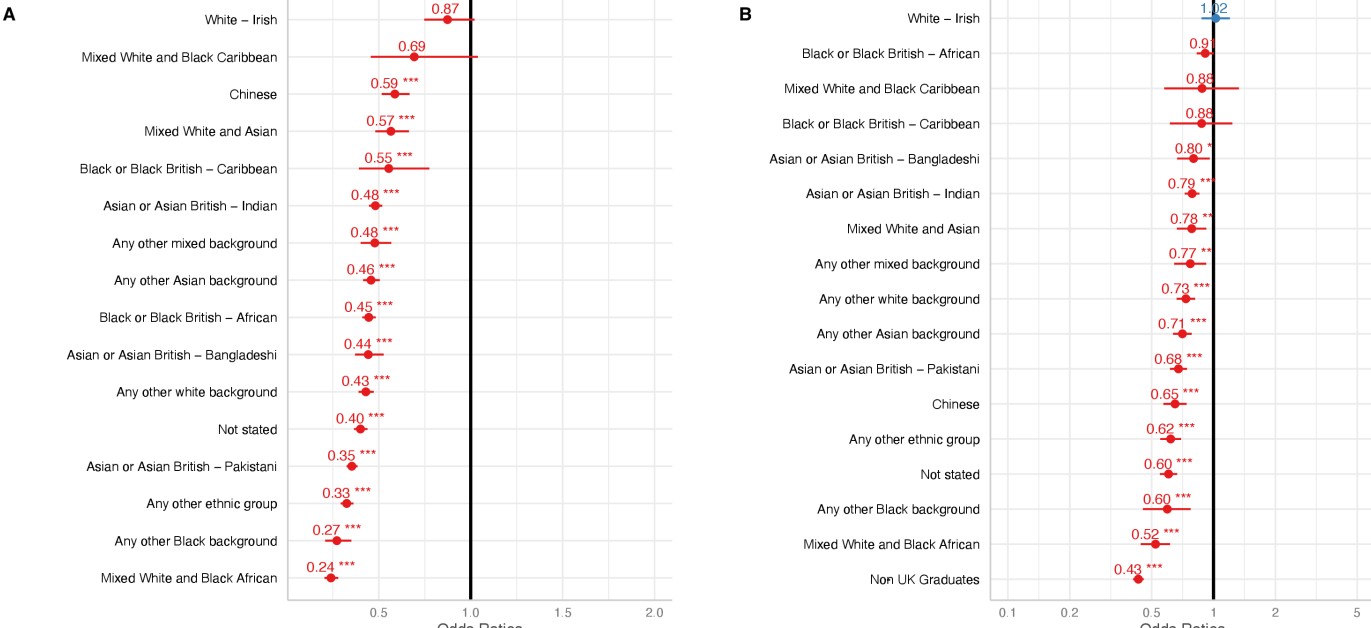

**Figure 4** Applications and outcomes for specialty training posts in the 2021–2022 recruitment year by ethnicity and graduate status. (A) Unadjusted ORs for success by ethnicity when compared with white British applicants derived from a logistic regression model (B) Adjusted ORs for success by ethnicity compared with white British where country of qualification (UK vs non-UK) is included as a covariate derived from a multivariable logistic regression model. Non-medical applicants to public health specialty training have been removed (n=988). ORs are presented with 95% CIs. *p<0.05, **p<0.01, ***p<0.001.

are not disproportionately lower or that this group is not disadvantaged at other stages of their career progression. Finally, to reiterate, the NHS workforce should reflect the population it serves at all levels. This extends beyond clinicians—similar work should be undertaken for allied health professionals and support staff in the NHS.

### Implications and supporting evidence

Over-representation of males in senior positions of medicine in the NHS, and in particular specialties such as surgery, has been noted previously[19] but national data on recruitment across specialty posts is limited. Success by female applicants in many specialties, demonstrated in our study, is a positive step towards gender balance, and perhaps reflects existing efforts to address disparities, or higher applicant quality where female applications are low. Additionally, data provided by NHS Digital demonstrate a steady rise in the proportion of female specialty trainees, from 45% in 2009 to 53% in 2017, suggesting either an increase in female applicants or successes, or a combination.[19] This does not, however, provide clarity on the distribution of female specialist trainees across specialties. The skew in applications and subsequent recruitment by gender, particularly among surgical specialties, is concerning. There is a clear attraction issue (ie, a lack of diversity among applicants) among multiple specialties (figure 2A and online supplemental figure 1), with several possible reasons. Taking surgery as an example, factors including a male-dominated workplace culture, incidents of bullying and harassment, few female role models and career inflexibility, have been suggested as the reasons why females are deterred from considering

to apply, as early as undergraduate study.[9 20] Female surgeons have reported quality of life and fewer unsocial hours as explanations of why women prefer other clinical specialties, in addition to the fear that working less than full time (LTFT) or taking career breaks is perceived negatively.[20] There are exceptions to this where some specialties appear to have succeeded in creating a culture that attracts female trainees and allows them to thrive. Obstetrics and gynaecology provide a clear example of a demanding specialty requiring shift work and a surgical skill set[21] where females dominate applications and successful hires. Obstetrics and gynaecology, public health, and paediatric medicine, are known to have a higher representation of females compared with males,[6] and continue to disproportionately recruit females; these represent three of the four specialties with the highest proportion of LTFT trainees.[22] The availability of LTFT opportunities compound gender segregation within specialties, and can exacerbate the gender pay gap.[6] Additional specialty-specific factors will inevitably influence a doctor's application decision-making, including personal choice and even patient preference (studies have demonstrated a slight preference for female obstetric and gynaecology clinicians by their female patient cohort)[23], and these should be taken into consideration. However, the aim should be to ensure that doctors are given the opportunity to flourish in all specialties and that this is facilitated by fair recruitment.

Concerningly, we find disparities in the success of applicants by ethnicity and this is not explained by country of graduation alone. A previous descriptive analysis

**Box 1   Actions for consideration to improve inclusivity in the specialty post recruitment process**

**Recruitment specific recommendations**
⇒ Processes and data pertaining to the allocation of specialty training posts should be transparent and published.
⇒ Ensure that recruitment panels are as diverse as possible.
⇒ Ensure that all staff involved in recruitment processes are trained in fair and inclusive recruitment, with regular refresher training.
⇒ Language used in applications should be inclusive and examined to ensure it does not deter any demographic group.
⇒ Review job criteria to ensure that only 'essential' skills and experience is included. Women, for example, are less likely to apply for a job where they do not meet all the criteria.
⇒ Include a self-identification category for gender.
⇒ Recruitment systems and processes should be flexible to accommodate for reasonable adjustments. This should be plainly messaged to applicants, along with a clearly signposted process for requesting reasonable adjustments.
⇒ Review application tests for inclusivity and fairness across protected characteristics.
⇒ Implement positive action schemes to support applicants from groups with disproportionately low success rates.

**Wider approaches to promote inclusivity to address attraction issues**
⇒ Support all genders with less-than-full-time applications.
⇒ Actively support women returning from maternity leave to increase confidence and mitigate deskilling (such as return to work transition programmes and 'keeping-in-touch' days).
⇒ Destigmatise parental leave for all genders.
⇒ Encourage male applications to traditionally female-dominated specialties to help break-down existing gender stereotypes within the National Health Service.
⇒ Promote diversity in specialties that are under-represented by certain demographic groups by highlighting role models and positive messaging.
⇒ Investigate cultural issues within specialties where this is a problem; carry out engagement surveys and focus groups to determine how these can be addressed.
⇒ Create and promote a culture with zero tolerance for bullying and harassment, and ensure that doctors feel safe to speak out.
⇒ Create outreach programmes for those specialties with attraction issues for certain groups.
⇒ Ensure workplaces are accessible and processes for reasonable adjustments (including successful examples of reasonable adjustments) are well advertised.
⇒ Engage with employee networks and patient groups to understand specific issues faced by different demographic groups and promote equality, diversity and inclusion-related activities.

demonstrated that doctors from minority ethnic groups were less likely to be considered suitable for appointment to specialty training posts in the UK, but did not consider country of graduation or undertake a thorough analysis of the data.[24] Examining combined infection training specifically, country of qualification possibly accounted for imbalances in recruitment by ethnicity.[13] Our findings, however, indicate a need to thoroughly review recruitment policies and processes from a diversity and inclusion perspective (such as bias in recruitment decisions

and a lack of inclusivity in the design of recruitment processes), to ensure that they are facilitating equitable outcomes. Outside of recruitment to specialty training, unequal progression by ethnicity has been demonstrated in the NHS. Doctors from minority ethnic groups are under-represented at consultant grade and within academic roles.[7] They have reported disproportionately high levels of discrimination from work colleagues, and had poorer outcomes at revalidation and Annual Review of Competence Progression, regardless of country of qualification. Additionally, differences in postgraduate examination pass rates by country of qualification, and between minority ethnic groups are established.[14] These differences can be overcome; data from NHS Digital demonstrates an incremental rise in the proportion of consultant grade doctors from minority ethnic groups over time, from 29% in 2009 to 36% in 2017,[19] while the NHS Workforce Race Equality Standard (WRES) demonstrated that in regions where a concerted effort to improve disparate outcomes for ethnic minority employees had been made, significant improvements were seen.[25] We should be mindful that unequal outcomes may reflect a history of disparities in the opportunities available to, and experiences of, various demographic groups and we encourage the NHS to support access schemes that aim to help doctors at an early-career stage. The 'widening access' initiative introduced by HEE, for example, is a positive step in attempting to support the applications of non-UK graduate doctors to specialty training posts[26]; schemes such as these should be maintained after HEE merges with NHS England. The fragmented nature of the NHS and its complex employment structures mean monitoring progression and attainment by protected characteristics is perhaps more complex than other professions, but effort must be made to formulate a consistent methodology to do so. The NHS is facing a workforce crisis[3] and doctors from ethnic minority backgrounds are being increasingly relied on in the UK. They comprise 41.9% (53 157) of the medical and dental workforce in NHS trusts and clinical commissioning groups in England in 2020[7]; therefore, ensuring that these doctors are able to work within an inclusive environment, that allows them to thrive and progress should be a priority.

Only 1.2% of the applicants to specialty training posts reported a disability—this will likely be an underestimate with under-reporting of disabilities among medical professionals.[27] It is encouraging, however, to observe a high proportion of acceptances among all individuals disclosing a disability, with a particularly high proportion of success seen in psychiatry and paediatric medicine. The 2021 Workforce Disability Equality Standard found 23.4% of the disabled staff working in the NHS did not have the adjustments necessary to effectively work in the NHS.[27] Ultimately, the NHS needs to ensure that application and recruitment processes are accessible and open to adjustments for all disabilities (including neurodiversity), eliminate any perceived fear of discrimination, and provide assurance that all NHS workplaces

will accommodate reasonable adjustments to ensure that disabled doctors can carry out their work. This will not only help to encourage more disabled applicants, but also make disabled clinicians to feel more comfortable disclosing this information, in turn support the monitoring and evaluation of diversity data. Additionally, given the diversity of issues faced by disabled doctors, we encourage assessment of this protected characteristic by taking the nature of one's disability into account. Stakeholders should engage with disabled doctors through networks such as the disability doctors network, to identify possible barriers to disclosing their disability, applying to become and working as a doctor, and how these could be overcome. Disabled doctors harbour unique insights and provide an opportunity to relate to the patient journey[28]; the NHS would stand to benefit from this expertise if they are supported to prosper within its environment.

## CONCLUSIONS

Overall, we present detailed analysis of the diversity and inclusion considerations for recruitment to specialty training in the UK for the 2021–2022 cycle. This study provides evidence to demonstrate higher success rates by females compared with men but gender segregation of applications by specialty. Our data highlight poorer outcomes for minority ethnic applicants, regardless of country of graduation. We show favourable outcomes for disabled doctors and encourage data reporting in this domain to be improved. Examination of recruitment processes is the first step towards building an inclusive workplace; we encourage policy-makers to investigate the root cause of our findings and ensure such monitoring is made routine.

**Acknowledgements** We thank members of the Freedom of Information office at Health Education England for providing data. We thank Mr Vivek Roy Chowdhury, Department of Economics, University of Cambridge, and Dr Oliver Feng, Department of Mathematics, University of Cambridge, for providing direction on statistical analysis.

**Contributors** Content guarantor: DA; Conceptualisation: DA and MR-C; Project administration: DA; Investigation: DA; Data curation: DA; Methodology: DA., MR-C; Visualisation: DA; Formal analysis: DA; Writing—original draft preparation: DA; Writing—review and editing: DA, MR-C, NX, SP; Funding acquisition: N/A; Supervision: SP.

**Funding** DA is a Clinical PhD Fellow and gratefully supported by the Wellcome Trust (grant number: 222903/Z/21/Z).

**Competing interests** None declared. Employers and/or funders had no role in study design, data collection and analysis, decision to publish, or preparation of the manuscript.

**Patient and public involvement** Patients and/or the public were not involved in the design, or conduct, or reporting, or dissemination plans of this research.

**Patient consent for publication** Not applicable.

**Provenance and peer review** Not commissioned; externally peer reviewed.

**Data availability statement** All data relevant to the study are included in the article or uploaded as online supplemental information.

**ORCID iD**
Dinesh Aggarwal http://orcid.org/0000-0002-5938-8172

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
