## [Reviewer comments · BMJ Open]

ARTICLE DETAILS

TITLE (PROVISIONAL)	Applications to medical and surgical specialist training in the UK National Health Service, 2021-22: a cross-sectional observational study to characterise the diversity of successful applicants
AUTHORS	Aggarwal, Dinesh; Roy-Chowdhury, Meera; Xiang, Nicola; Peacock, Sharon

VERSION 1 – REVIEW

REVIEWER	Subramaniam , J. Calvary Health Care Adelaide
REVIEW RETURNED	17-Dec-2022

GENERAL COMMENTS	I highly commend this research and the use of FOI requests to acquire the large dataset that made it possible. In my own opinion, the data paints both a reassuring and worrying picture (higher success of females vs 'attraction' issues in certain specialities, overall success rate of disabled vs non-disabled candidates), but the authors are clear that this is a snapshot of recruitment, and the message that this data should be routinely collected, analysed and reported remains an important one. While I have made a few comments above, I would hold this publication ready for print and will be closely following the final print version and comments that follow. Recommendation: Comments: Thank you for the opportunity to review this manuscript. I note the extensive comments that have come before me, and the significant work that you have undertaken in the peer review process to date. My comments are outlined below: Outline: 'Examining diversity and inclusion in recruitment to medical and surgical specialist training in the UK National Health Service, 2021-22: an observational study' includes an analysis of data from the 2021-22 recruitment into speciality training posts, and comments on differences in rates of applications made and successful recruitment by protected characteristics (gender, ethnicity, disability). The paper outlines a higher proportion of male candidates for most surgical specialities, and a higher proportion of female candidates for obstetrics and gynaecology. There is an overall higher proportion of successful candidates across all specialities. Disabled applicants seem to be more successful than their non-disabled counterparts.
---

	The paper highlights the importance of this kind of data to be routinely collected and analysed in future. Objectives and rationale: The objectives of the paper are clearly stated and I agree that they are incredibly important in the future of NHS recruitment. Replicability and reproducibility: As the data are gathered via FOI, the study is both replicable and reproducible. Statistical analyses, sampling and reporting: The statistical analysis is robust and sampling is as thorough as it can be given the limitations of FOI requests. The data are reported in detail. Tables/figures: Data are displayed exhaustively in the provided charts and tables. The breakdown by speciality is clearly displayed Interpretation of data: Interpretation of the data is clear and well explained. Strengths and weaknesses of study, methods and arguments: The fact that country of qualification has been included in analysis is a notable strength when compared to similar work previously conducted. The large sample size allows for generalisability, and the fact that this work is a snapshot of a single year of recruitment highlights the importance of repeated reporting of this data. These strengths have all been noted by both the authors and other peer reviewers. Limitations of study, methods and arguments: I wonder if the data across multiple years could be pooled via FOI request in order to gain numbers >5 and therefore the ability to catch a greater proportion of candidates? While I hope that this research will prompt more regular collection and reporting of this data, should there be a need to repeat this work via journal publication, this may be a suggestion for future work. The other potential weaknesses remain in the limitations of the data collected, including self reported gender and the lack of diversity in options, which lie outside the control of the authors. This has been commented on and appropriate recommendations are made. Structure, flow or writing: The paper is well written and reads well Language editing: No further language editing is required Other comments:
--	---

	I highly commend this research and the use of FOI requests to acquire the large dataset that made it possible. In my own opinion, the data paints both a reassuring and worrying picture (higher success of females vs 'attraction' issues in certain specialities, overall success rate of disabled vs non-disabled candidates), but the authors are clear that this is a snapshot of recruitment, and the message that this data should be routinely collected, analysed and reported remains an important one. While I have made a few comments above, I would hold this publication ready for print and will be closely following the final print version and comments that follow.
--	---

REVIEWER	Wang, Andrew UCLA
REVIEW RETURNED	27-Dec-2022

GENERAL COMMENTS	The authors should be commended on their work in examining diversity and inclusion across specialties in the UK. With minor revisions, I recommend the paper be accepted for publication. Congratulations on a well-done study! I have no comments for the abstract, introduction, stats, references, table/figures, or supplementary sections, as the authors have addressed the prior reviewers' comments satisfactorily. I have two primary comments: My biggest concern is the title is a bit misleading and should match the conclusion of characterizing successful applicants from diverse backgrounds. As it stands, the title of the paper suggests more of a diversity catalog or improvement in diversity recruitment trends over time which is not the case and not necessarily the same thing. To add to that, while I realize that the paper is the first of its kind to comprehensively catalog this topic, it would be significantly stronger and more impactful if they were able to establish a "baseline %" from a previous dataset (as opposed to simply referencing papers from specific specialties for comparison) of what the break down of diverse successful candidates were in (as an example) 2010 and 2000. By doing so, I would feel more confident regarding statements that suggest "diversity is improving but needs more work". My other suggestions are minor (below) and center on improving the readability and clarity of the paper: 129 Methods section, Statistical analysis: It should be made clear if authors will be denoting trends or "favors" with statistically significant or not. As currently written, the results suggest anything with a noted p-value is significant, otherwise, it is just a "trend". Please clarify. 129 Methods section, the type of disability is not discussed. If not provided or unable to assess, please state here not just in the discussion. 172 "the break down of these groups". Please add (e.g. surgical) or something to that effect as it primes the reader for the results section of what categories were created. 187 Results section, would the study benefit from the primary care
---

	subgroup analysis vs specialty care? This is a hot topic in the US, but may not be in UK. If that is included in the analysis via GP then state which specialties that would include. For example, in the US its Peds,IM, FM, medpeds, obgyn 198, 224, 225, 245 “in favor” “more successful”, does that mean trended to but not statistically significant or? See Methods section 129 comment. 201, 218 formally, the first character of a sentence should not be an algebraic number unless spelled out. 201-206, 229, 246 In other sections, you report proportions with percentiles. This is not the case here and should be corrected for consistency. 203 “excluding applicants with undisclosed gender” should be stated in the methods section not results. 213-214 Please clarify the sentence 229-230, missing numerical values of OR 233-234 this statement, while discussing a result, is not a result. Hence should belong in the methods 267 a limitation statement should include subgroup specialty analyses that were not all performed due to limitations in sample size/power between groups. If this is not true, then more data should be provided 267 there should be a limitation statement suggesting a reason or need for trend analysis of previous years 306, similar to comment for 267, while the authors do characterize the % of success for females, the authors utilize other studies to demonstrate baseline, which is a weaker form of evidence
--	--

VERSION 1 – AUTHOR RESPONSE

Reviewer: 1

Dr. J. Subramaniam , Calvary Health Care Adelaide

Recommendation:

Comments:

Thank you for the opportunity to review this manuscript. I note the extensive comments that have come before me, and the significant work that you have undertaken in the peer review process to date. My comments are outlined below:

Outline:

'Examining diversity and inclusion in recruitment to medical and surgical specialist training in the UK National Health Service, 2021-22: an observational study' includes an analysis of data from the 2021-22 recruitment into speciality training posts, and comments on differences in rates of applications made and successful recruitment by protected characteristics (gender, ethnicity, disability). The paper outlines a higher proportion of male candidates for most surgical specialities, and a higher proportion

of female candidates for obstetrics and gynaecology. There is an overall higher proportion of successful candidates across all specialities. Disabled applicants seem to be more successful than their non-disabled counterparts. The paper highlights the importance of this kind of data to be routinely collected and analysed in future.

We thank Reviewer 1 for this succinct summary of the work.

Objectives and rationale:

The objectives of the paper are clearly stated and I agree that they are incredibly important in the future of NHS recruitment.

We thank Reviewer 1 for underlining the importance of the objectives of this work to specialty trainee recruitment in the NHS.

Replicability and reproducibility:

As the data are gathered via FOI, the study is both replicable and reproducible.

Thank you for highlighting this.

Statistical analyses, sampling and reporting:

The statistical analysis is robust and sampling is as thorough as it can be given the limitations of FOI requests. The data are reported in detail.

We thank Reviewer 1 for highlighting the robustness of the analysis undertaken and we agree the work has been carried out within the limitations of FOI requests.

Tables/figures:

Data are displayed exhaustively in the provided charts and tables. The breakdown by speciality is clearly displayed

Thank you.

Interpretation of data:

Interpretation of the data is clear and well explained.

Once again, we thank Reviewer 1 for this comment.

Strengths and weaknesses of study, methods and arguments:

The fact that country of qualification has been included in analysis is a notable strength when compared to similar work previously conducted. The large sample size allows for generalisability, and the fact that this work is a snapshot of a single year of recruitment highlights the importance of repeated reporting of this data. These strengths have all been noted by both the authors and other peer reviewers.

We agree with comments made here – particularly with the strength of the manuscript lying in the use of country of qualification as a co-variate and the large dataset. We stress the need for such analyses to be routine and reported to address possible barriers for under-represented demographic groups, and in turn to maximise the inclusivity and success of the NHS specialty recruitment process.

Limitations of study, methods and arguments:

I wonder if the data across multiple years could be pooled via FOI request in order to gain numbers >5 and therefore the ability to catch a greater proportion of candidates? While I hope that this research will prompt more regular collection and reporting of this data, should there be a need to repeat this work via journal publication, this may be a suggestion for future work. The other potential weaknesses remain in the limitations of the data collected, including self reported gender and the lack

of diversity in options, which lie outside the control of the authors. This has been commented on and appropriate recommendations are made.

We thank Reviewer 1 for highlighting this important point. Unfortunately, we have been informed Health Education England do not retain applicant records for more than one year at a time, and therefore seeking data for a longitudinal analysis is not possible. The large dataset analysed in this study does, however, allow the capture of the vast majority of specialties (lines 207-208, "56/58 Specialty Training Posts had complete data representing male and female applications and 40/58 Specialty Training Posts had complete data representing successful male and female applications (excluding applicants with undisclosed gender", and all ethnic groups had sufficient applicants and acceptances to satisfy the requirements for data release).

Structure, flow or writing:

The paper is well written and reads well

Thank you.

Language editing:

No further language editing is required

Thank you.

Other comments:

I highly commend this research and the use of FOI requests to acquire the large dataset that made it possible. In my own opinion, the data paints both a reassuring and worrying picture (higher success of females vs 'attraction' issues in certain specialities, overall success rate of disabled vs non-disabled candidates), but the authors are clear that this is a snapshot of recruitment, and the message that this data should be routinely collected, analysed and reported remains an important one.

While I have made a few comments above, I would hold this publication ready for print and will be closely following the final print version and comments that follow.

We are very grateful for this positive description of the important work presented. We agree that the work is limited by the fact it is a cross-sectional analysis, and as mentioned, we highlight this in the limitations section. We appreciate the comments relating to the interest and impact this manuscript will generate.

Reviewer: 2

Dr. Andrew Wang, UCLA

Comments to the Author:

The authors should be commended on their work in examining diversity and inclusion across specialties in the UK. With minor revisions, I recommend the paper be accepted for publication. Congratulations on a well-done study!

We thank Reviewer 2 for their positive remarks on the study and we hope to have sufficiently addressed the comments made.

I have no comments for the abstract, introduction, stats, references, table/figures, or supplementary sections, as the authors have addressed the prior reviewers' comments satisfactorily. I have two primary comments:

My biggest concern is the title is a bit misleading and should match the conclusion of characterizing successful applicants from diverse backgrounds. As it stands, the title of the paper suggests more of a diversity catalog or improvement in diversity recruitment trends over time which is not the case and not necessarily the same thing.

We thank Reviewer 2 for this comment and agree that we can provide more clarity in our title. We have therefore changed it to, *“Applications to medical and surgical specialist training in the UK National Health Service, 2021-22: a cross-sectional observational study to characterise the diversity of successful applicants”*.

To add to that, while I realize that the paper is the first of its kind to comprehensively catalog this topic, it would be significantly stronger and more impactful if they were able to establish a “baseline %” from a previous dataset (as opposed to simply referencing papers from specific specialties for comparison) of what the break down of diverse successful candidates were in (as an example) 2010 and 2000. By doing so, I would feel more confident regarding statements that suggest “diversity is improving but needs more work”.

We thank Reviewer 2 for this important point. Establishing such a baseline would be very useful but as mentioned above, unfortunately, historical data on applications and successes to specialist training is not retained beyond a year by Health Education England (the source organisation for this information). We do agree that providing some helpful historical context, where available, does enhance the manuscript and have therefore provided information on changes in the proportion of women in Specialist Registrar level post (line 323), and Consultant level for ethnicity (as the most relevant data available), for the years 2007 to 2017 (line 367).

My other suggestions are minor (below) and center on improving the readability and clarity of the paper:

129 Methods section, Statistical analysis: It should be made clear if authors will be denoting trends or “favors” with statistically significant or not. As currently written, the results suggest anything with a noted p-value is significant, otherwise, it is just a “trend”. Please clarify.

We thank Reviewer 2 for pointing this out. The use of the term ‘in favour of’ is purely to allow readers to infer the demographic group with a greater success (particularly in the pairwise comparisons of percentage of successful candidates) and we do not state trends in the results. The results, for differences in proportions in a pairwise comparison, are presented as 95% confidence intervals with the implication that if the confidence interval does not cross the line of no-effect, it can be taken as statistically significant. We have now stated this in the methods for clarity (lines 175-178)

129 Methods section, the type of disability is not discussed. If not provided or unable to assess, please state here not just in the discussion.

Thank you, data for the category of disability is unfortunately not available and has been addressed in previous response to reviews. We have however additionally stated this in the methods lines 155-156, as suggested by Reviewer 2.

172 “the break down of these groups”. Please add (e.g. surgical) or something to that effect as it primes the reader for the results section of what categories were created.

We thank Reviewer 2 for this suggestion and have provided additional wording in line 173.

187 Results section, would the study benefit from the primary care subgroup analysis vs specialty care? This is a hot topic in the US, but may not be in UK. If that is included in the analysis via GP then state which specialties that would include. For example, in the US its Peds, IM, FM, medped, obgyn
The purpose of this work is to examine the diversity of applications and outcomes across the specialty trainee recruitment in the UK National Health Service, as a whole. We have focused on examining the entire cohort and after significant additional analysis, as requested by the BMJ statistical editor, illustrate 95% confidence intervals for differences in success to specialty training for all individual specialties by gender (including primary care, i.e. General Practice in the UK). We have not carried out any further comparative analysis (e.g. primary care vs specialty care), as we envisage the work presented will be of equal interest to all specialties and stakeholders. Further, small numbers at specialty level would result in an underpowered study from which conclusions would be less robust. Though outside of the scope of this work, we have made all raw data available for further analyses, if of interest to readers.

198, 224, 225, 245 “in favor” “more successful”, does that mean trended to but not statistically significant or? See Methods section 129 comment.

Please see comment above – we have clarified the interpretation of statistical significance inferred

from 95% confidence intervals in the methods section. The use of the term 'in favour of' is purely to allow readers to infer the demographic group with a greater success (particularly in the pairwise comparisons of percentage of successful candidates) and not to suggest a trend, which is also visible from the results as presented. We have changed the text of "more successful" to "significantly more successful" in line 231, and direct readers to Supplementary Table 4 for further details.

201, 218 formally, the first character of a sentence should not be an algebraic number unless spelled out.

Thank you for pointing this out, this has now been corrected throughout the manuscript.

201-206, 229, 246 In other sections, you report proportions with percentiles. This is not the case here and should be corrected for consistency.

Thank you, we have corrected this throughout the manuscript.

203 "excluding applicants with undisclosed gender" should be stated in the methods section not results.

Thank you, we have corrected this in the manuscript (lines 179-180).

213-214 Please clarify the sentence

Thank you, we have corrected this in the manuscript to, "The gender balance of successful recruits to specialities largely reflected that of the pool of applicants (Figure 2c)".

229-230, missing numerical values of OR

The text avoids duplication of data presentation and states 11/15 minority ethnic groups had a lower odds ratio of success at recruitment when compared to White-British applicants. Readers are then directed to Figure 4b and Supplementary Table 5, where the values can be visualised and read.

233-234 this statement, while discussing a result, is not a result. Hence should belong in the methods We thank Reviewer 2 for highlighting this. This sentence refers to additional results available in Supplementary Tables 5 and 6, and was inserted as per the request of the statistical editor at the BMJ.

267 a limitation statement should include subgroup specialty analyses that were not all performed due to limitations in sample size/power between groups. If this is not true, then more data should be provided

We thank Reviewer 2 for making this point. We have provided significant re-analysis at individual specialty level previously in line with the statistical editor's request at the BMJ. We do provide a descriptive comparative analysis of application proportions and successes by gender and demonstrate segregation of recruitment. Further comparative analysis between specialities would not only be beyond the aims and scope of this study but conclusions would also be limited by the small numbers at each specialty level. Additionally, data for ethnicity and disability have a greater amount of missing data at specialty level. As suggested, we have highlighted this in the limitations section (lines 286-288). We do provide all raw data and direct readers to this in the manuscript (line 166-167).

267 there should be a limitation statement suggesting a reason or need for trend analysis of previous years

We thank Reviewer 2 for this comment. We do provide a statement acknowledging the need for a longitudinal analysis (lines 74, 293-297, 414, and Table 1). As stated above, we have been informed historic data is not available from Health Education England. As pointed out by previous reviewers and Reviewer 1, we hope this work will encourage stakeholders to collect and analyse data pertaining to diversity and inclusion more routinely, given the clear benefits of doing so (as presented in the manuscript).

306, similar to comment for 267, while the authors do characterize the % of success for females, the authors utilize other studies to demonstrate baseline, which is a weaker form of evidence

Thank you. We agree with Reviewer 2 that this is indeed an issue – to our knowledge, diversity and inclusion data on recruitment to specialist training in the UK are lacking in the public domain – and as highlighted above are not available via Freedom of Information requests. We have, in line with Reviewer 2's suggestion above, provided further evidence on baseline trends in numbers of female specialist trainees (line 323) and consultant's by ethnicity (line 367) over time. We utilise available

data and studies in our introduction and discussion to draw attention to the equality, diversity, and inclusion related issues faced by doctors in the NHS. Accordingly, improving data collection is highlighted as an important conclusion in the manuscript.

VERSION 2 – REVIEW

REVIEWER	Subramaniam , J. Calvary Health Care Adelaide
REVIEW RETURNED	28-Feb-2023
GENERAL COMMENTS	none further As I have requested no changes, I have no further comments for peer review.